# Text Summarization Method Based on Gated Attention Graph Neural Network

**DOI:** 10.3390/s23031654

**Published:** 2023-02-02

**Authors:** Jingui Huang, Wenya Wu, Jingyi Li, Shengchun Wang

**Affiliations:** College of Information Science and Engineering, Hunan Normal University, Changsha 410081, China

**Keywords:** encoder-decoder, GNN, contrastive learning, confidence calculation of important sentences, attention mechanism

## Abstract

Text summarization is an information compression technology to extract important information from long text, which has become a challenging research direction in the field of natural language processing. At present, the text summary model based on deep learning has shown good results, but how to more effectively model the relationship between words, more accurately extract feature information and eliminate redundant information is still a problem of concern. This paper proposes a graph neural network model GA-GNN based on gated attention, which effectively improves the accuracy and readability of text summarization. First, the words are encoded using a concatenated sentence encoder to generate a deeper vector containing local and global semantic information. Secondly, the ability to extract key information features is improved by using gated attention units to eliminate local irrelevant information. Finally, the loss function is optimized from the three aspects of contrastive learning, confidence calculation of important sentences, and graph feature extraction to improve the robustness of the model. Experimental validation was conducted on a CNN/Daily Mail dataset and MR dataset, and the results showed that the model in this paper outperformed existing methods.

## 1. Introduction

Text summarization is an information compression technique that converts a collection of text or documents into a short summary that covers the important information in the original text with little redundant information and high readability [1,2]. Nowadays, the mainstream text summarization techniques are divided into two main categories: extractive summarization and generative summarization. Extractive abstracts [3,4] are formed by extracting important sentences from the original document, ranking them in order of importance, and then forming the abstract. This method is simple to implement and has natural advantages in content selection and sentence internal structure coherence, but it has low coherence when extracting important sentences to form summaries. Generative abstracts [5,6] can generate sentences or phrases that are not in the original text, making the sentences flow more smoothly with each other and greatly improving the coherence and readability of the abstract.

At present, the text summary model based on deep learning has achieved remarkable results in the task of summary generation. Most of the existing models use an encoder-decoder framework, where the encoder encodes the input into intermediate states and the decoder processes the intermediate states and then outputs them. Among them, Sequence-to-Sequence (Seq2Seq) models [7,8,9] using RNN, Bi-GRU, Bi-LSTM, etc. are widely used as encoders for generative text summarization tasks and have achieved some good results. However, the model has a very strong semantic dependency and may be semantically incomplete in the decoding stage. Refs. [10,11,12,13] proposed that the attention mechanism is introduced on top of the original model, which alleviates the limitations of the model to some extent, but tasks such as text summarization require richer semantic features. Ref. [14] made an excellent contribution to extracting more semantic features by constructing a stacked network structure SRCLA to filter different types of features. With the proposed document-level tasks, the single-layer coding and decoding framework cannot meet the demand, so a layered coding framework is proposed. Currently, a hierarchical codec framework has become the mainstream framework for text summary tasks, which can encode and decode sentences and documents, respectively, in the encoding stage. This effectively alleviates the problem of document-level tasks, but it still cannot make full use of input sequence information. As Graph Neural Networks (GNN) [15,16,17] can fully extract the information in the corresponding graph structure without destroying the semantics of the sentence, it effectively improves the effectiveness of text summary generation. Yao et al. [18] constructed charts by graph convolutional networks (GCN) to analyze the overall information of a document, which effectively improved text classification. To improve the correlation between the original document and the abstract, Ma et al. [19] introduced a gated attention encoder into the model. Lin et al. [20] used gated convolutional neural networks to deal with word repetition and semantic irrelevance. Shi et al. [21] used GRU gated recurrent units and a self-attentive layer to extract more semantic features and improve the interpretability of the text classification model. The introduction of gated units can effectively improve the performance of the model. Recently, many researchers have focused their research attention on gated graph neural network models that consider both text sequence structure information and graph structure information, and have achieved remarkable results in text summarization tasks. However, this requires a large amount of time to be reserved for data processing problems. To address this problem, a text summarization model combining Gated Graph Neural Networks (GGNN) [22,23,24,25] with attention is proposed, and the model has better performance than most baseline models. To improve the robustness of the graph structure, refs. [26,27,28,29,30] introduced contrast learning in the model for pulling positive samples close to push away negative samples. Most of the encoders of existing models are based on sentences or words with low reliability, and the joint summary model proposed in [31] improves the performance of summary generation.

Most of the above models ignore the influence of local fine irrelevant information on the accuracy of summary generation. To address the problem, this paper proposes a gated attention graph neural network model framework. The main innovative elements include the following:To adequately exclude irrelevant information, an attention gate is added to the gate control unit GRU.In the iterative process of the existing Gated Graph Neural Network (GGNN), irrelevant information will also be accumulated and amplified, resulting in redundant information in the decoding phase which cannot be eliminated, making the text abstract distorted. Therefore, an Attention Gate (GA) is added to the gating unit GRU to form a Gate Attention Graph Neural Network (GA-GNN) model.Use of parallelism in the coding phase to mitigate inadequate coding and high time complexity.If a single coder is used in the sentence encoding stage, the local information and the global information cannot be concerned at the same time, which easily leads to insufficient semantic information encoding. Tandem encoding can solve the problem of global local semantic encoding, but the time complexity is high. In this paper, parallel sentence coding mode is used to encode both local and global information of the text, which enriches vector information and shortens training time.A joint loss function optimization model is proposed.Decoding based on graph-extracted features not only ignores the connections between sentence levels, but also diminishes the accuracy of generated summaries due to the lack of multiple sample guidance. In this paper, the loss function is optimized by weights based on contrast learning, graphic feature extraction and confidence calculation of important sentences, and all the key information is effectively incorporated into the decoder.

## 2. Related Technologies

### 2.1. Codec Framework

The codec framework is an architecture method for machine translation and natural language processing, which mainly includes two parts: encoder and decoder. The encoder accepts the input sequence and encodes the entire sequence information, eventually mapping the input to a hidden state containing intermediate semantics. The decoder converts the original sequence to the target sequence using the intermediate semantic representation and output of the decoder. The codec framework can be Recurrent Neural Network (RNN) and its variant networks (such as LSTM, Bi-LSTM), Convolutional Neural Network (CNN), Attention mechanism and Transformer, etc.

With the increasing difficulty of natural language processing tasks, single-layer encoders are gradually unable to meet the requirements, so layered encoders are proposed, which have now become the mainstream framework for text summarization tasks. The encoding model is mainly divided into two parts: sentence encoding and document encoding. Sentence encoding is to input the word vector representations corresponding to all words contained in each sentence of the original document into the network to generate the corresponding hidden-level sentence vector representations. Document encoding is the use of neural networks to generate document-level sentence vectors from the hidden layer sentence vector representations.

### 2.2. Graph Neural Network (GNN)

The traditional neural network model mainly focuses on the sequence-related information and cannot express the non-sequence-related information in a friendly way. Graph is a kind of non-sequential structure, but its edges and nodes can effectively represent the dependency relationship between each other, so Graph Neural Network (GNN) was born. Because of its good learning ability and interpretation ability, it has become a widespread graph analysis model.

The vertices in the graph represent the words and the edges represent the relationships between the vertices. The purpose is to obtain the feature representation of each node in the graph by learning the neighboring nodes. Early graph neural networks update node-like by iteratively exchanging information of neighboring nodes, but the problem of cyclic layer interdependence tends to occur during the iterative process, so variants of graph neural networks such as Graph Convolution Networks (GCN), Graph Attention Networks (GAN) and Gated Graph Neural Networks (GGN) have been derived.

Gated Graph Neural Network (GGNN) is a GRU-based spatial domain model that solves the long-term text dependency problem through a gating mechanism. Each sentence in the document generates a graph structure, the words in the sentence are considered as graph nodes, and the edges between the nodes represent relational features. The main purpose of the GGNN model is to update the node information through the gating mechanism, so that the updated node can contain its neighboring node information. The GGNN model takes the neighbor information of the node as the input and its own state information as the hidden state, controls the generation of new information by updating the gate, resets the forgetting of the gate control information, calculates the newly generated information by using the tanh() function, and finally controls the balance state between the forgetting degree of the hidden state at the last moment and the input degree of the newly generated information by updating the gate.

### 2.3. Contrastive Learning

Contrastive learning, which was first applied to CVs, has shown great potential in natural language processing in recent years. The core of contrast learning is to learn the difference between samples by constructing positive and negative samples. The specific idea is to compare a given sample with its semantically similar samples (positive samples) and semantically dissimilar samples (negative samples) by constructing a contrast loss function, so that the semantically similar samples are represented more closely in spatial distance and the semantically dissimilar samples are further in spatial distance, thus achieving a similar clustering effect.

## 3. Model

In the text summary generation task, the source text contains a large amount of redundant information, which is not only time-consuming but also of poor summary generation quality if it cannot be removed during the training process. This paper proposes a neural network model GA-GNN based on the gated attention graph, as shown in Figure 1. The model structure mainly consists of three parts: encoder, gated graph neural network, and decoder. Firstly, the input text is encoded into a sentence encoding vector and a document vector containing global and local semantic information using a hierarchical encoder. Secondly, the sentence vectors are converted into independent graph data structures and fed into GA-GNN to update the feature information of the nodes and reduce the accumulation of local irrelevant information during the iteration process. Finally, the LSTM-ATT network with attention is used for decoding to complete the generation of a text summary. Multiple loss functions were combined to optimize the model.

### 3.1. The Hierarchical Encoder

The layered encoder consists of sentence encoding and document encoding. Bi LSTM and CNN parallel networks are used for sentence encoding, and only Bi LSTM networks are used for document encoding. Bi-LSTM can preserve future and past information through a double-layer LSTM, capable of learning a more adequate and accurate global semantic representation of words. CNN is an excellent network for learning local semantic information, which can represent local semantic information more accurately than other networks.

Given a sentence $s=\left\{x_{1},x_{2},\cdots ,x_{n}\right\}$, the sentence length is *n*, $x_{i}$ represents the *i*-th word, and word embedding is expressed as $W=\left\{w_{1},w_{2},\cdots ,w_{n}\right\}$; the word is embedded into the input parallel network. The encoding process is shown in Formulas (1)–(3):(1)$$h^{B}=Bi\_LSTM\left(x_{t},h_{t-1}\right)$$
(2)$$h^{c}=CNN\left(w⨂k+b\right)$$
(3)$$h^{S}=\left[\left[h_{1}^{B},h_{1}^{C}\right],\left[h_{2}^{B},h_{2}^{C}\right],\cdots ,\left[h_{n}^{B},h_{n}^{C}\right]\right]=\left[h_{1}^{S},h_{2}^{S},\cdots ,h_{n}^{S}\right],$$
where $h^{B}$ and $h^{C}$ represent the coding vector generated by Bi-LSTM and CNN encoder, respectively, ⨂ denotes the convolution operation, *k* denotes the convolution kernel size, and $h^{S}$ denotes the sentence encoding vector containing global semantic and local semantic information generated by combining the two encoding vectors by the same dimensional merging method.

To further sense the correlation between adjacent sentences, the sentence encoding vector h is input into the document encoder Bi-LSTM, and the hidden layer states learned before and after are stitched together as the document encoding vector $h_{i}^{D}$, which is calculated as shown in (4).
(4)$$h_{t}^{D}=Bi\_LSTM\left(h_{t}^{S},h_{t-1}\right)$$

### 3.2. Gated Attention Feature Extraction

In order to make full use of the graph structure information, eliminate irrelevant information, and allow each node to better fuse the neighboring information, a graph neural network incorporating the Gate recurrent Attention Unit (GAU) is introduced to make the nodes perform information fusion as comprehensively as possible and obtain a more advanced feature representation.

Firstly, the text graph G = (V, E, A) is established according to the relationship between words, where V represents the set of nodes. Each independent word is regarded as a node, E represents the set of all edges, and the edges between nodes correspond to the relationship between words. The structured information of the graph G is stored in the adjacency matrix $A\in R^{D\left|V\right|\times 2D}$, consisting of the feature set of the incoming and outgoing edge information of the nodes in the graph. At moment *t*, the gated attention unit takes as input the result of the interaction between the word node and its adjacency matrix $a_{v}^{t}$, which is calculated as follows.
(5)$$a_{v}^{t}=A_{v:}^{T}{\left[h_{1}^{\left(t-1\right)T},\;\cdots ,h_{\left|V\right|}^{\left(t-1\right)T}\right]}^{T}+b,$$
where |*V*| represents the number of nodes, and $A_{v:}$ is the corresponding row vector of node *v* in the adjacency matrix; $\left[h_{1}^{\left(t-1\right)T},\;\cdots ,h_{\left|V\right|}^{\left(t-1\right)T}\right]$ is the state matrix of all nodes at moment *t* − 1, and *b* denotes the bias.

Then, feature extraction is performed by gating the attention unit GAU, which is a key component of the GA-GNN model and is formed by adding an attention gate based on GRU, whose structure is shown in Figure 2. The calculation methods of the reset gate $r_{v}^{t}$ and update gate $z_{v}^{t}$ in the original GRU remains unchanged.
(6)$$r_{v}^{t}=\sigma \left(W_{r}a_{v}^{t}+U_{r}h_{v}^{t-1}+b_{r}\right),$$
(7)$$z_{v}^{t}=\sigma \left(W_{z}a_{v}^{t}+U_{z}h_{v}^{t-1}+b_{z}\right),$$
(8)$${\tilde{h}}_{v}^{t}=tanh\left(W_{\tilde{h}}a_{v}^{t}+U_{r}\left(r_{v}^{t}*h_{v}^{t-1}+b_{\tilde{h}}\right)\right),$$
where *σ* stands for sigmoid function, *W* and *U* stand for weight parameter matrix, and *b* stands for bias parameter.

To prevent the accumulation and amplification of small local irrelevant information and to ensure the accuracy of feature extraction, an attention gate GA is added to explicitly calculate the degree of influence between any nodes during each update iteration to eliminate the local information in the input $a_{v}^{t}$ that is irrelevant to the current node. The output of the attention gate GA is ${\hat{h}}_{v}^{t}$ defined as:(9)$${\hat{h}}_{v}^{t}=\sum_{v=1}^{n}a_{v}^{t}*softmax\left(a_{v}^{t}\right),$$

Finally, the output $h_{v}^{t}$ of neighbor information node is obtained by fusing the gated attention unit GAU.
(10)$$h_{v}^{t}=\left(1-z_{v}^{t}\right)*h_{v}^{t-1}+z_{v}^{t}*{\tilde{h}}_{v}^{t}+{\hat{h}}_{v}^{t},$$

### 3.3. Contrastive Loss Function

The key of contrastive learning is to construct a contrast loss function. First, the positive and negative samples are obtained based on the data transformation; then, the feature space is compared by calculating the similarity of the two samples, and finally the positive samples are approximated and the negative samples are pushed away.
(11)$$L_{con}=-\frac{1}{\left|N\right|}\log\frac{e^{sim\left(h_{i},h_{i}^{+}\right)/\tau }}{e^{sim\left(h_{i},h_{j}^{+}\right)/\tau }+\sum_{j=1}^{K}\left(e^{sim\left(h_{i},h_{j}^{-}\right)/\tau }\right)},$$
where |*N*| is the number of samples, $h_{i}$ is the original sample, $h_{i}^{+}$ denotes a positive sample and $h_{j}^{-}$ denotes a negative sample. The numerator represents the cosine similarity between the original sample and the positive sample, that is, the distance between the two vectors. The denominator represents the dot product of the original sample to all positive and negative samples. *K* is the number of negative samples. When all the remaining data in the original sample are taken as negative samples, the computational complexity is high, so a part of the data is selected as negative samples by sampling. *τ* is the temperature parameter, set to 0.1, $sim\left(h_{i},h_{i}^{+}\right)$ and $sim\left(h_{i},h_{j}^{-}\right)$ are the cosine similarities between samples and positive and negative samples. For each positive sample, the similarity between a single positive sample and the original sample is calculated by the following formula:(12)$$sim\left(h_{i},h_{i}^{+}\right)=\frac{h_{i}h_{i}^{+}}{\left|\left|h_{i}\right|\right|\cdot \left|\left|h_{i}^{+}\right|\right|},$$
where $h_{i}$, $h_{i}^{+}$ are the feature representations of the samples.

### 3.4. Confidence Calculation of Important Sentences

There is a large amount of redundant information in the source text that is not related to the central idea of the text, and filtering this irrelevant information before generating the abstract can effectively improve the accuracy of the abstract. In this model, the document level coding vector is used as the input of the important sentence confidence calculation module, and the sentences are extracted based on the calculation results.
(13)$$P_{i}=softmax\left(\left(W*f_{i}\right)\right),$$
where *softmax* () is the activation function, *W* is the parameter matrix, * is the matrix product, and $f_{i}$ represents the result of Dropout dimension reduction for the *i*-th sentence in the document. Calculated according to the following formula:(14)$$f_{i}=Dropout\left(Relu\left(W*h_{i}^{D}+b\right)\right),$$

Finally, the important sentences are extracted based on the magnitude of the calculation result *P_i_*.

The loss function (denoted as $L_{D}$) for the confidence calculation of the significant sentence is calculated by the following formula.
(15)$$L_{D}=-\frac{1}{M}\sum_{i\in M}{\hat{y}}_{i}logp\left({\hat{y}}_{i}|w\right)+\left(1-{\hat{y}}_{i}\right)logp\left({\hat{y}}_{i}|w\right),$$
where *M* denotes the total number of training sentences, ${\hat{y}}_{i}$ denotes the label, when ${\hat{y}}_{i}=0$ means the *i*-th sentence is irrelevant and is not used as a summary sentence, and when ${\hat{y}}_{i}=1$, the *i*-th sentence should be used as a summary sentence.

### 3.5. Decoders and Loss Functions

In order to predict the target result, the LSTM network with attention mechanism is used for decoding. At moment *t*, the hidden state $z_{t-1}$ generated by the decoder for the word at the moment *t* − 1, the vector representation $h_{t-1}$ of the subword in the summary at moment *t* − 1, generated by the gated attention graph neural network, and the context vector $c_{t-1}$ are input to the decoder to convert the hidden layer state $z_{t}$ at the current moment.
(16)$$z_{t}=LSTM\left(h_{t-1},z_{t-1},c_{t-1}\right),$$

The context vector $c_{t}$ is calculated as follows:(17)$$c_{t}=\sum_{j=1}^{n}\frac{\exp\left(u_{t,j}\right)}{\sum_{k=1}^{n}\exp\left(u_{t,k}\right)}h_{j},$$
where $u_{t,j}=\tan h\left(Wz_{t-1}+Uh_{j}+b\right)$, $z_{t-1}$ denotes the hidden state of the decoder at moment *t* − 1, $h_{j}$ denotes the feature vector of the *j*th word after passing through the GA-GNN network, *W* and *U* denote the weight parameter matrix, and *b* denotes the bias parameter.

In order to effectively use the key information in the decoding stage, combined with the contrast learning loss function $L_{con}$ and the important sentence confidence calculation loss function $L_{D}$, the loss function of the gated attention feature extraction model was extended as follows:(18)$$L=-\frac{1}{N}\sum_{t\in T}logp(Z|S;\theta )+\lambda L_{D}+\beta L_{con},$$
where *S* is the sequence of input sentences, *Z* is the sequence of words in the abstract, and *N* is the total number of trained words. *λ* and *β* represent the weight of loss terms $L_{D}$ and $L_{con}$, respectively, which are determined as learnable parameters in training.

## 4. Experimental Results and Analysis

### 4.1. Dataset

This paper adopts the CNN/Daily Mail dataset (a news dataset composed of a CNN dataset and a Daily Mail dataset, It contains 280,940 training sets, 13,368 verification sets, 11,490 test sets), and MR Dataset (movie review document dataset containing 5331 positive documents and 5331 negative documents, the first 4000 as the training set and 1331 reviews as the verification set (test set), respectively).

### 4.2. Parameter Setting and Evaluation Index

In this experiment, word embedding size and hidden layer state vector dimension were both set to 521 during model training. Adam method was used to optimize the model, the learning rate was set to 0.001, and the batch size was set to 100.

Evaluation indexes are used to measure the quality of the model. The evaluation indexes of the text summary are divided into manual evaluation and automatic evaluation. The manual evaluation will be affected by the subjective tendency of the evaluator. In the experiment, three indexes of Rouge-1, Rouge-2 and Rouge-L in ROUGE, a standard automatic text summary evaluation method based on recall rate proposed by Lin, were used as the evaluation criteria of the model.

One-tuple evaluation indicators:(19)$$Rouge\_1=\frac{\sum_{s\in \left\{Ref\right\}}\sum_{gram1\in S}Count_{match}\left(gram1\right)}{\sum_{S\in \left\{Ref\right\}}\sum_{gram1\in S}Count\left(gram1\right)}$$

Binary evaluation indicators:(20)$$Rouge\_2=\frac{\sum_{s\in \left\{Ref\right\}}\sum_{gram2\in S}Count_{match}\left(gram2\right)}{\sum_{S\in \left\{Ref\right\}}\sum_{gram2\in S}Count\left(gram2\right)}$$
where N_gram denotes the nthword, {Ref} denotes the standard summary, the numerator counts the total number of simultaneous occurrences of N_gram in the generated summary and the standard summary, and the denominator counts the number of occurrences of monomials and binary groups in the standard summary.
(21)$$R_{lcs}=\frac{LCS\left(X,Y\right)}{len\left(m\right)},$$
(22)$$P_{lcs}=\frac{LCS\left(X,Y\right)}{len\left(n\right)},$$
(23)$$Rouge\_L=\frac{\left(1+\beta^{2}\right)R_{lcs}P_{lcs}}{R_{lcs}+\beta^{2}P_{lcs}},$$
where *L* is the longest common subsequence, *X* and *Y* denote the standard summary and generated summary, $R_{lcs}$ and $P_{lcs}$ denote the recall and accuracy of the longest common subsequence.

### 4.3. Ablation Experiments

In order to verify the impact of each module proposed in this paper on the accuracy of summary generation, relevant ablation experiments are conducted and the optimal model is selected by comparing the performance results. The CNN/Daily Mail dataset is selected for the following ablation experiments.

From Table 1, it can be seen that the performance of the model decreases to some extent by removing the important sentence confidence calculation, the comparison learning module and the gated graph attention network; from the results of the decreasing data of the three modules, it can be seen that GA-GNN has a more significant impact on the performance of the model, from which it can be concluded that GA-GNN plays a decisive role in the process of text summary generation. The comparison of the results from Experiment 3 and Experiment 7 shows that the experimental effect of introducing attention gates to the model in the gating unit is superior, proving that the gated attention unit can eliminate more redundant information in the process of updating parameters, thus substantially improving the utilization of important information and laying a good foundation for the generation of the final text summary. It can be seen from experiments 5, 6 and 7 that when the encoder uses a single network, the model effect is lower than that of the multi network encoder model. The reason is that the vector encoded by a single network cannot make full use of the local and global key information of the text, with the result that the final text summary cannot accurately represent the core content of the original text. The performance of the encoder based on a serial network is not significantly different from that of the encoder model based on parallel network. However, during the experiment, the running time of the sentence encoder based on a serial network is about 4 days, while the running time of the model based on a parallel sentence encoder is about 2.5 days. The time complexity of the parallel network is less than that of the serial network. Therefore, this paper selects the parallel network as the sentence encoder for coding.

Figure 3 shows the effects of different modules in the model. According to the experimental results of the model on the CNN/Daily Mail dataset, the results of the GA module on the three indicators are higher than those of other modules. This shows that the GA module is a key factor affecting the model performance, and the combination of all modules will greatly improve the model performance.

### 4.4. Baseline Model Comparison Experiment

In order to verify the validity of the models, the following four baseline models are selected as the subjects of comparison experiments.
Seq2Seq + Joint Attention (2018): Hou Liwei et al. [32] proposed to incorporate a joint attention mechanism into the decoder to reduce redundant repetitive information in the decoding process by the decoder.DAPT (2022): Li et al. [33] proposed a dual-attention pointer fusion network fusing contextual and critical information.AGGNN (2022): Deng et al. [24] proposed an attention-based gated graph neural network that effectively exploits the semantic features of words.GRETEL (2022): Qianqian Xie et al. [29] introduced a graphical contrast topic augmented language model in the model, and combined the graphical contrast topic model with the training model to fully capture the global semantic information.
CNN/Daily Mail DatasetMR Dataset

From the above experimental data in Table 2 and Table 3, we can see that the model proposed in this paper has improved compared with other comparative models in three indicators. Compared with GRETEL, the indicators Rouge-1, Rouge-2 and Rouge-L in the data set CNN/Daily Mail increased by 1.48%, 1.26% and 0.44%, respectively. The data set MR has been improved by 1.69%, 0.66% and 1.02%, respectively, which shows that the model can fully understand the context semantic information and integrate the key information of the source text, so as to generate a higher quality and more coherent text summary.

## 5. Conclusions

In this paper, a framework model based on gated attention graph neural network is proposed for generating more concise and fluent text summaries. This model uses a parallel mechanism to encode sentences to make full use of local and global semantic information in the text. After that, sentence vectors are converted into graph structure information for feature extraction. However, due to the accumulation and amplification of local redundant information in the extraction process, it will be impossible to eliminate them in the coding stage, affecting the accuracy and simplicity of summary generation. Therefore, gated attention units are introduced into the model, which are used in combination with graph neural networks to eliminate redundant information and improve model performance. In order to effectively incorporate key information into the decoding process, this paper proceeds to optimize the model loss function in three parts: contrast learning, important sentence confidence calculation and graph neural network to generate more reasonable and accurate text summaries. The effectiveness of the model framework proposed in this paper is demonstrated by conducting experiments on two standard general-purpose datasets. In future work, we will continue to explore how to optimize the model framework and establish the relationship between graph nodes and edges to obtain better results.

## Figures and Tables

**Figure 1 sensors-23-01654-f001:**
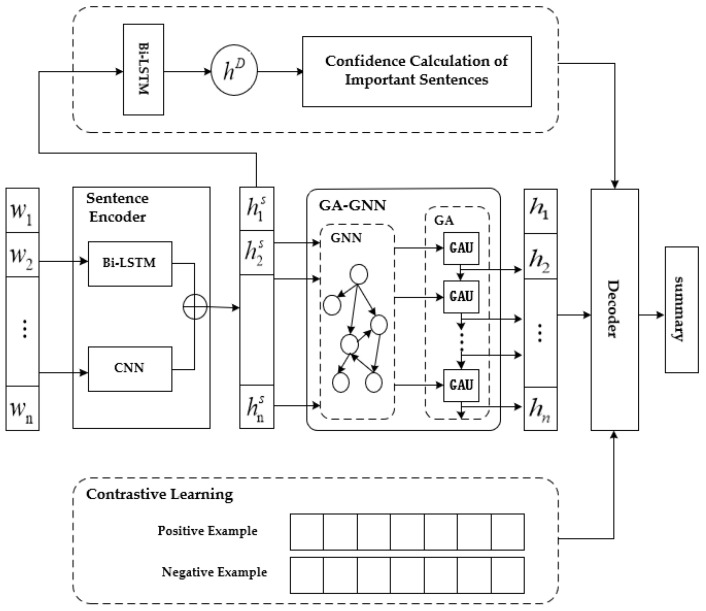
Model framework based on gated graph attention network.

**Figure 2 sensors-23-01654-f002:**
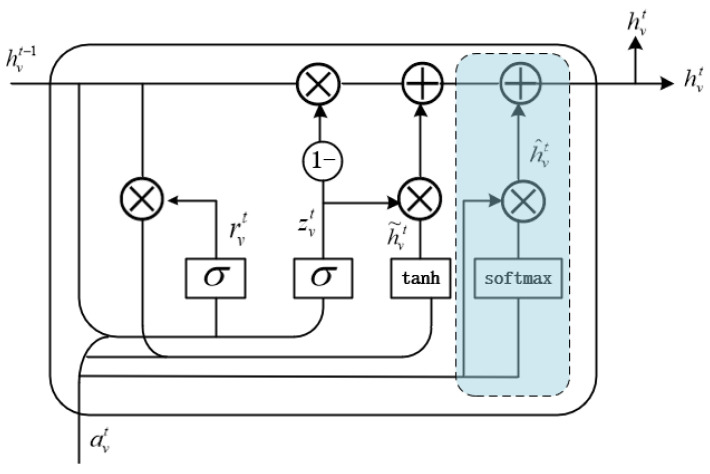
Gated Attention Unit GAU.

**Figure 3 sensors-23-01654-f003:**
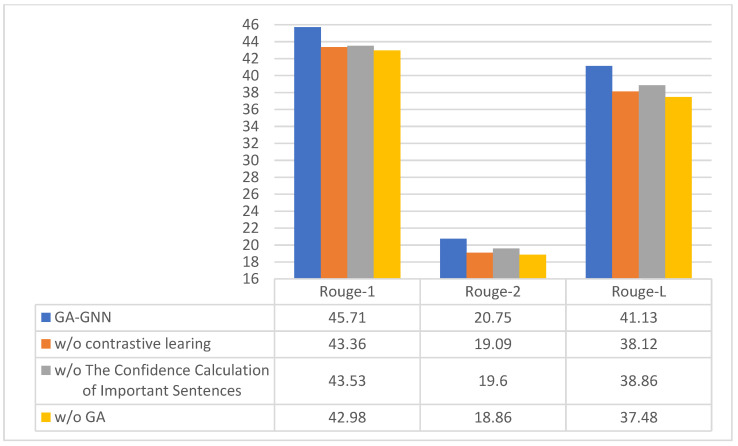
Experimental results of the effect of different modules on model performance.

**Table 1 sensors-23-01654-t001:** Ablation experiments.

Number	Confidence Calculation of Important Sentences	Contrastive Learning	GA	GNN	Sentence Encoder	Rouge-1	Rouge-2	Rouge-L
1	×	√	√	√	parallel connection	43.53	19.60	38.86
2	√	×	√	√	parallel connection	43.36	19.09	38.12
3	√	√	×	√	parallel connection	42.98	18.86	37.48
4	√	√	×	×	parallel connection	42.77	18.28	37.36
5	√	√	√	√	Single network	44.13	19.68	39.48
6	√	√	√	√	series connection	44.81	20.48	39.23
7	√	√	√	√	parallel connection	**45.14**	**20.75**	**41.13**

**Table 2 sensors-23-01654-t002:** Evaluation results of the baseline model on the CNN/Daily Mail dataset (%).

Model	Rouge-1	Rouge-2	Rouge-L
Seq2Seq + Joint Attention	27.80	14.25	25.71
DAPT	40.72	18.28	37.35
AGGNN	42.25	19.13	38.65
GRETEL	43.66	19.46	40.69
Our Method	**45.14**	**20.75**	**41.13**

**Table 3 sensors-23-01654-t003:** Evaluation results of the baseline model on the MR dataset (%).

Model	Rouge-1	Rouge-2	Rouge-L
Seq2Seq + Joint Attention	38.55	17.36	36.38
DAPT	39.27	17.56	36.13
AGGNN	40.68	18.10	37.54
GRETEL	43.02	20.19	38.53
Our Method	**44.71**	**20.85**	**39.55**

## Data Availability

Not applicable.

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
