# Peer review of "Text Summarization Method Based on Gated Attention Graph Neural Network"

_sensors, 2023, doi:10.3390/s23031654_

Round 1

Reviewer 1 Report

This paper proposed a graph neural network model GA-GNN based on gated attention, which effectively improved the accuracy and readability of text summarization. Here are a few questions:

1. inappropriate citation of references and lack of international authoritative literature. This affects the comprehensive and cutting-edge nature of the literature.

A relevant latest reference can be concerned, such as

Two End-to-End Quantum-inspired Deep Neural Networks for Text Classification, IEEE Transactions on Knowledge and Data Engineering, doi: 10.1109/TKDE.2021.3130598 (2021)

2.
I could not appreciate what exactly this method has done in Table 2 and Table 3 drawn by the authors, and the indicators where the method of this paper has worked need to be highlighted.

3. The full text feels like a solution to the text summary problem using the method of someone else. What is the specific theoretical or experimental original work that needs to be highlighted more.

Reviewer 2 Report

Authors addressed the method for enhancing summarizaion tasks.

The idea on GRU is sound, Also, the contrastive learning should be interested for journal readers.

My recommendations are as followed.

1. The concept of contrastive loss function is a littile ambiguous in Eq. (11). I am confused since there is operatior in Eq. (11), which is prime ( ' ) in the summation of negative samples. Authors should address on the prime operator.

2. The similarity function in Eq. (12), there is another prime operator ( ' ). I recommend authors more description in Eq. (12) as well as Eq. (11)

Reviewer 3 Report

 An deep learning algorithm, graph neural network model GA-GNN based on gated attention has been employed in this manuscript in order to improves the accuracy and readability of text summarization. authors stated that the results showed that the model in this paper outperformed existing methods. The manuscript has been well written, and has good contributions in their research fields.

 My question is, is the topic of the manuscript FIT for the “sensors” Journal? As the aims of the Journal is “Sensors (ISSN 1424-8220) provides an advanced forum for the science and technology of SENSOR and its applications”. I don’t think the manuscript fit for this Journal.

Round 2

Reviewer 1 Report

The authors have addressed all my questions.

Reviewer 3 Report

The Response letter is reasonable